# Immunopathogenesis of ANCA-Associated Vasculitis

**DOI:** 10.3390/ijms21197319

**Published:** 2020-10-03

**Authors:** Andreas Kronbichler, Keum Hwa Lee, Sara Denicolò, Daeun Choi, Hyojeong Lee, Donghyun Ahn, Kang Hyun Kim, Ji Han Lee, HyungTae Kim, Minha Hwang, Sun Wook Jung, Changjun Lee, Hojune Lee, Haejune Sung, Dongkyu Lee, Jaehyuk Hwang, Sohee Kim, Injae Hwang, Do Young Kim, Hyung Jun Kim, Geonjae Cho, Yunryoung Cho, Dongil Kim, Minje Choi, Junhye Park, Junseong Park, Kalthoum Tizaoui, Han Li, Lee Smith, Ai Koyanagi, Louis Jacob, Philipp Gauckler, Jae Il Shin

**Affiliations:** 1Department of Internal Medicine IV (Nephrology and Hypertension), Medical University Innsbruck, 6020 Innsbruck, Austria; andreas.kronbichler@i-med.ac.at (A.K.); sara.denicolo@i-med.ac.at (S.D.); philipp.gauckler@i-med.ac.at (P.G.); 2Department of Pediatrics, Yonsei University College of Medicine, Seoul 03722, Korea; AZSAGM@yuhs.ac; 3Yonsei University College of Medicine, Yonsei University, Seoul 03722, Korea; julie715@naver.com (D.C.); annie302@naver.com (H.L.); afduhn75@yonsei.ac.kr (D.A.); khkpaul@yonsei.ac.kr (K.H.K.); jihanlee@yonsei.ac.kr (J.H.L.); kht2006@yonsei.ac.kr (H.K.); minha312312@yonsei.ac.kr (M.H.); jsw915@naver.com (S.W.J.); cjblue6821@yonsei.ac.kr (C.L.); hojunenate@gmail.com (H.L.); haejune.sung@yonsei.ac.kr (H.S.); rabbit960902@naver.com (D.L.); pskyline@yonsei.ac.kr (J.H.); ucb.sohee@gmail.com (S.K.); joshuahwang@yonsei.ac.kr (I.H.); dyaiden@naver.com (D.Y.K.); maeng806@yonsei.ac.kr (H.J.K.); matia125@yonsei.ac.kr (G.C.); sophie137@naver.com (Y.C.); snoopy4276@naver.com (D.K.); ariraf2@yonsei.ac.kr (M.C.); jhjh524@yonsei.ac.kr (J.P.); vinc@yonsei.ac.kr (J.P.); 4Department of Basic Sciences, Division of Histology and Immunology, Faculty of Medicine Tunis, Tunis El Manar University, Tunis 1068, Tunisia; kalttizaoui@gmail.com; 5University of Florida College of Medicine, University of Florida, Gainesville, FL 32610, USA; lih2@ufl.edu; 6The Cambridge Centre for Sport and Exercise Science, Anglia Ruskin University, Cambridge CB1 1PT, UK; lee.smith@anglia.ac.uk; 7Research and Development Unit, Parc Sanitari Sant Joan de Déu, CIBERSAM, 08830 Barcelona, Spain; a.koyanagi@pssjd.org (A.K.); louis.jacob.contacts@gmail.com (L.J.); 8ICREA, Pg. LluisCompanys 23, 08010 Barcelona, Spain; 9Faculty of Medicine, University of Versailles Saint-Quentin-en-Yvelines, 78000 Versailles, France

**Keywords:** ANCA, biomarker, phenotype, treatment, pathogenesis

## Abstract

Anti-neutrophil cytoplasmic antibody (ANCA)-associated vasculitis is an autoimmune disorder which affects small- and, to a lesser degree, medium-sized vessels. ANCA-associated vasculitis encompasses three disease phenotypes: granulomatosis with polyangiitis (GPA), microscopic polyangiitis (MPA), and eosinophilic granulomatosis with polyangiitis (EGPA). This classification is largely based on clinical presentations and has several limitations. Recent research provided evidence that genetic background, risk of relapse, prognosis, and co-morbidities are more closely related to the ANCA serotype, proteinase 3 (PR3)-ANCA and myeloperoxidase (MPO)-ANCA, compared to the disease phenotypes GPA or MPA. This finding has been extended to the investigation of biomarkers predicting disease activity, which again more closely relate to the ANCA serotype. Discoveries related to the immunopathogenesis translated into clinical practice as targeted therapies are on the rise. This review will summarize the current understanding of the immunopathogenesis of ANCA-associated vasculitis and the interplay between ANCA serotype and proposed disease biomarkers and illustrate how the extending knowledge of the immunopathogenesis will likely translate into development of a personalized medicine approach in the management of ANCA-associated vasculitis.

## 1. Introduction

Anti-neutrophil cytoplasmic antibody (ANCA)-associated vasculitis is a systemic autoimmune disease, predominantly affecting small and medium-sized vessels (i.e., capillaries, venules, arterioles, and small arteries). ANCA-associated vasculitis comprises three distinct disease phenotypes, granulomatosis with polyangiitis (GPA, formerly Wegener’s granulomatosis), microscopic polyangiitis (MPA), and eosinophilic granulomatosis with polyangiitis (EGPA) [1].

ANCA play an important role in the pathogenesis of ANCA-associated vasculitis, which is summarized in Figure 1. The seminal work by van der Woude et al. in 1985 found circulating antibodies in patients with GPA, which correlated with disease activity [2]. Depending on their immunofluorescence pattern on ethanol-fixed neutrophils, ANCA were termed cytoplasmic (c-ANCA), perinuclear (p-ANCA), and atypical ANCA [3,4]. In line with the progress made during the past decades in our understanding of ANCA-associated vasculitis, screening methods have been improved and high-quality immunoassays aimed at the detection of the ANCA serotype, namely proteinase 3 (PR3)-ANCA and myeloperoxidase (MPO)-ANCA, are recommended according to a recent international consensus statement [5]. ANCA with an epitope specificity for two major antigens, MPO and PR3 in the cytoplasm of neutrophils, is present in most cases with severe disease presentations and underlying GPA or MPA, while only half of the patients with localized forms of GPA test positive for ANCA. 

In general, PR3-ANCA is associated with GPA, while MPO-ANCA is predominantly found in cases with MPA; however, overlap exists and cases with MPO-ANCA positive GPA and PR3-ANCA positive MPA are a focus of recent research [6,7]. Cases with double seropositivity for both PR3- and MPO-ANCA are rarely seen, and are mainly associated with secondary forms of ANCA-associated vasculitis (i.e., in cocaine-induced forms or drug-induced vasculitis) [8,9]. Recent research found that the ANCA serotype better discriminates between genetic associations, therapeutic response, relapse risk, prognosis and co-morbidities (venous thromboembolic events and cardiovascular death) than a classification based on the clinical phenotype [10,11,12,13,14,15,16]. In addition, the respective ANCA serotype more closely relates to biomarkers of disease activity [17] and may help to identify novel therapeutic targets and predict response to current treatment regimens. Thus, this review will focus on delineating the immunological aspects of ANCA-associated vasculitis with a focus on the ANCA serotype classification system. EGPA is underrepresented in biomarker studies and significantly differs from the other entities, and thus is not represented in this work.

## 2. Overview of ANCA-Associated Vasculitis

### 2.1. Genetic and Epigenetic Changes in ANCA-Associated Vasculitis

The exact mechanisms leading to an excess production of ANCA are not clear. In healthy individuals, PR3- and MPO-ANCA are detected in circulation [18]. These findings argue that further steps are necessary for the onset of autoimmunity. A variety of different factors have been identified for their implication in the pathogenesis of ANCA-associated vasculitis, including the environment, genetics, and infections [19]. In brief, genome-wide association studies (GWAS) have revealed that PR3-ANCA vasculitis is associated with the genes *SERPINA1* (encoding alpha-1 antitrypsin), *PRTN* (encoding PR3), and in line with other autoimmune diseases, human leukocyte antigen (HLA) loci, including *HLA-DP4*, while MPO-ANCA vasculitis has been reported to be associated with *HLA-DQ* [10,20]. Most of the associations correlated stronger (76%) with the ANCA serotype than with the clinical phenotype [21]. Epigenetic factors including low DNA methylation in regulating MPO and PR3 transcription have also been found to be associated with ANCA-associated vasculitis [22]. Thus, evidence suggests that both genetic and epigenetic factors are involved in the complex pathogenesis of ANCA-associated vasculitis. 

### 2.2. Pathogenesis of ANCA-Associated Vasculitis

Kessenbrock et al. showed that neutrophil extracellular traps (NETs) are released by ANCA-stimulated neutrophils and contain the respective target antigens PR3 and MPO. They further demonstrated that the deposition of NETs in affected organ systems contributes to the damage observed in cases with ANCA-associated vasculitis [23]. NET components are capable of activating dendritic cells and in turn inducing autoimmunity, leading to ANCA production [24,25].

Neutrophils are the most important effector cells in the pathogenesis of ANCA-associated vasculitis. In the normal human immune system, neutrophils act as a first line defense system through engulfment of external pathogens, degranulation of anti-microbials, and formation of NETs. However, under certain conditions, resting neutrophils in the bloodstream go through a process known as “priming”, whereby neutrophils display target antigens (e.g., MPO or PR3) on their surface membranes [26,27]. Priming may be caused by a number of processes, including treatment-related reactions, infections, and activation of the alternative complement pathway. Concomitant “hits” such as infections, silica exposure or drugs can induce such a reaction [9,19,28]. After detection of infectious agents, transforming growth factor (TGF)-beta and interleukin (IL)-6 released from dendritic cells induce differentiation of naive T cells into T helper 17 (Th17) cells [29]. Subsequently, IL-17 from Th17 stimulates macrophages to produce tumor necrosis factor (TNF)-α and IL-1ß, which are known to act as major priming factors [30]. In addition, activation of the alternative complement pathway results in generation of C5a [31], which can prime neutrophil activation by binding to the C5a receptor on neutrophils [32]. 

The initial mechanisms of ANCA generation are not well understood. Chronic nasal colonization with *Staphylococcus aureus* is associated with relapse in patients with an established diagnosis of GPA, and treatment with trimethoprim-sulfamethoxazole reduced the risk [33]. Treating peripheral blood mononuclear cells (PBMCs) from GPA patients with B cell activating factor (BAFF) and IL-21 increases ANCA production, which is further exacerbated with oligodeoxynucleotides containing CpG motifs, a pathogen-associated immunostimulant [34]. This suggests that hyperactivation of B cells and T cells is involved in initiation of the ANCA production. The exposed autoantigens interact with ANCA which results in excessive activation of neutrophils adhering to endothelial cells. This hyperactivation is followed by abnormal cytokine production and the release of reactive oxygen species (ROS) and lytic enzymes, which result in vascular endothelial cell injuries. Receptor-interacting protein kinase 1/3 (RIPK1/3)/ mixed-lineage kinase-like domain (MLKL)-dependent necroptosis induces the release of NETs, which scaffold the alternative complement pathway activation [35]. Moreover, exposure to NET components such as histones and matrix metalloproteinases induced by excessive activation of neutrophils are associated with vascular endothelial cell injuries [36,37,38,39]. PR3 has also been shown to reinforce vascular damage in vitro [9]. NET components themselves include PR3 and MPO, and chronic elevation of these enzymes in circulation leads to their recognition by dendritic cells and subsequently T cells and plasma cells as neoantigens [23,24,25]. The continuous production of PR3-ANCA and MPO-ANCA from lymphocytes results in a vicious cycle of neutrophil hyperactivation, inflammatory activity, and vasculitis. Thus, neutrophils, ANCA elevation, disruption of plasma and T cell tolerance, and overproduction and persistence of NETs together contribute to the pathogenesis of PR3-and MPO-ANCA vasculitis.

## 3. Classification of ANCA-Associated Vasculitis According to Clinical Phenotypes

### 3.1. Brief Description of the Different Phenotypes

The two disease phenotypes have distinct and common characteristics, which are briefly discussed. GPA is a necrotizing granulomatous inflammation usually involving the respiratory tract (most often the ear, nose and throat (ENT)-tract and to a lesser degree the lower respiratory tract) [40]. A high proportion of patients with systemic, generalized GPA test positive for ANCA, yet a small subset of GPA cases with ENT-limited disease, are ANCA-negative [41,42,43,44,45,46].

MPA is generally present without granulomatous inflammation and affected organs differ from cases with GPA [47]. MPA mainly affects lungs and kidneys, while other manifestations such as peripheral neuropathy or cardiac involvement seem to be underreported. Involvement of the eyes and the ENT tract is reported less frequently in these cases [47]. Given its systemic character, most patients are ANCA-positive and a positive test for MPO-ANCA is most frequently found. The clinical diagnoses MPA and GPA are often grouped together on the grounds of similarities in clinical presentation, especially when kidney involvement is present, and comparable histologic findings when a biopsy is performed (“pauci-immune” crescentic glomerulonephritis of the kidney). To date, the concept of a single disease spectrum led to the inclusion of both diseases in the same clinical trials and development of comparable therapeutic strategies to manage both diseases. 

Within GPA, there are few differences in clinical characteristics between patients positive for PR3- or MPO-ANCA. ANCA-negative GPA patients score lower on the Birmingham Vasculitis Activity Score compared to ANCA-positive GPA with a lower frequency of kidney involvement [6]. The degree of ANCA expression may also predict relapse in patients with kidney involvement, regardless of PR3- or MPO-ANCA serotype [48]. ANCA-negativity also predicts longer time to relapse in patients treated with rituximab, as shown by McClure et al. using a 57-patient sample, of which 37% had kidney involvement [49]. However, the value of ANCA-negativity in predicting relapse remains controversial and requires further study. PR3- and MPO-ANCA, their phenotypes, and their associated biomarkers are shown in Figure 2.

PR3-ANCA vasculitis is characterized by a predominant involvement of the upper respiratory tract, and in comparison to MPO-ANCA vasculitis less frequently affects the lower respiratory tract and the kidneys [47]. Both entities present with a similar kidney histology, while in PR3-ANCA vasculitis the number of normal glomeruli is higher and usually a lower amount of interstitial fibrosis characteristic for chronic damage is found, explaining the higher rate of kidney recovery in these patients [50,51]. Furthermore, endothelial PR3 internalization leads to apoptosis, while endothelial MPO internalization stimulates intracellular oxidant production [52,53]. Treatment of both MPO- and PR3-ANCA to date is similar, but more modern treatment strategies and differences in the pathophysiology between both entities and differences in presentation would suggest a more “tailored approach”, i.e., the tempo of kidney function decline. In MPO-ANCA vasculitis, an association between proteinuria and kidney outcome was proposed and these patients may benefit from RAS inhibitors [52].

### 3.2. Current Classification Criteria 

Early classification criteria of GPA have been developed by the American College of Rheumatology (ACR) in 1990, with the inclusion of 85 vasculitis cases, respectively [54], while no such criteria were published for MPA. Using the respective ACR criteria in a contemporary cohort of patients highlighted the need for an update, since sensitivity for diagnosing GPA was rather low (65.6%), while the specificity was 88.7% [55]. The Chapel Hill Consensus Conference (CHCC) issued a nomenclature system (nosology) of vasculitis in 1994 and sub-divided GPA and MPA [56]. In 2012, an update was issued, incorporating the revised name GPA (instead of Wegener’s granulomatosis), and subdividing small vessel vasculitides (SVV) into immune complex SVV and ANCA-associated SVV [1]. An algorithm by the European Medicines Agency (EMA) to implement a methodology for the classification of ANCA-associated vasculitides was proposed in 2007, which used the ACR criteria and the CHCC definitions, and incorporated surrogate markers for vasculitis and ANCA [57]. After revision of the CHCC in 2012, re-application of the EMA algorithm found an excellent kappa statistic of 0.96 with one patient changing from MPA to GPA and vice versa [58]. Updates in diagnostic and classification criteria are expected within subsequent years and will incorporate a set of clinical and laboratory specificities of these diseases, and may also take into account the more accurate separation by ANCA serotype than the actual clinical phenotype. Although classification criteria are currently based on the CHCC nomenclature and are set to change, we classified ANCA-associated vasculitides based on their serotype (PR3- or MPO-ANCA) or clinical presentation (GPA or MPA) to maintain consistency with referenced studies in this review.

## 4. Pathogenetic Steps in ANCA-Associated Vasculitis and ANCA Serotype Specificity

### 4.1. Regulation of Cytokines and Related Molecules According to ANCA Serotype

A summary of inflammatory molecules and their expression levels in ANCA-associated vasculitis classified based on the ANCA serotype (PR3- or MPO-ANCA) is shown in Table 1. Both T and B cells are critically involved in the pathogenesis of ANCA-associated vasculitis. T cells are found in vasculitic lesions and in granulomas. Circulating effector T cell populations are expanded and are in a persistent state of activation, while regulatory T cell subsets are impaired in their function. At the site of inflammation, T cells and dendritic cells are abundant and under the influence of several cytokines (Table 1) that orchestrate the immunological response [59]. T cells show persistent activation during phases of remission, while B cell activation is associated with disease activity [60]. In general, B cell homeostasis is perturbed during active phases of ANCA-associated vasculitis, with an expansion of cluster of differentiation (CD)38 and a decreased expression of CD5 [61]. Several lines of evidence highlight further involvement of B cells in the pathogenesis of ANCA-associated vasculitis, including B cell activating factor release by ANCA-activated neutrophils, ANCA epitope spreading to produce pathogenic antibodies, overexpression of ANCA autoantigen genes, and further steps eventually leading to B cell and plasma cell production of pathogenic ANCA [62]. The seminal studies Rituximab in ANCA-associated vasculitis (RAVE) [63,64] and Rituximab versus cyclophosphamide in ANCA-associated vasculitis (RITUXVAS) [65] have underlined the relevance of B cells, as targeting B cells by treatment with rituximab, a monoclonal antibody directed against CD20-bearing cells, has become mainstay in the management of PR3- and MPO-ANCA vasculitis. Again, pro-inflammatory cytokines and molecules regulate B cell maturation and finally activation. A summary of different candidates is highlighted in Table 1.

#### 4.1.1. Priming of Neutrophils and Monocytes as a Key Step in the Pathogenesis of PR3-and MPO-ANCA Vasculitis

One of the key steps in the pathogenesis of ANCA-associated vasculitis is priming of neutrophils, which in turn leads to expression of ANCA on the cell membrane. This process is triggered by systemic or tissue-specific proinflammatory stimuli. In both diseases (PR3-and MPO-ANCA vasculitis) several such stimuli as TNF-α [66,67,68], C5a [32], IL-1β [112], IL-2Rα (CD25) [74], IL-6 [71], IL-18 [72], granulocyte colony-stimulating factor (G-CSF), granulocyte-macrophage colony-stimulating factor (GM-CSF) [76], high-mobility-group-protein B1 (HMGB1), and macrophage migration inhibitory factor (MIF) are elevated in comparison to controls. Elevated levels of ADAM metallopeptidase domain 17 (ADAM17) and α1-trypsin polymers [80] have been reported in PR3-ANCA vasculitis, while the expression of ADAM17 in MPO-ANCA vasculitis has not been tested so far. In contrast, the expression of CD122 (IL-2Rβ) on CD4+ T cells [74] is reduced in ANCA-associated vasculitis. More recently, the role of monocytes in the complex pathogenesis was reinforced [113]. Increased expression levels of TNF-α, interferon (IFN)-γ [105,106] and ADAM17, also implicated in the priming of monocytes and in part indicative of Th1 involvement, were observed in PR3-ANCA vasculitis. 

#### 4.1.2. Activation of Neutrophils and Monocytes in PR3-and MPO-ANCA Vasculitis

As Kessenbrock et al. demonstrated, NETs released by ANCA-stimulated neutrophils play a critical role in pathways leading to vascular damage [23] and further ANCA production [24,25]. A variety of different stimuli are involved in neutrophil and monocyte activation as a further step perpetuating the vasculitic process. Among these stimuli, the involvement of the complement system as a systemic stimulus and monocyte chemoattractant protein-1 (MCP-1) at the site of inflammation emerged as centrally involved [75]. 

IL-8, one of the most important neutrophil chemotactic factors, can attract and activate neutrophils that can in turn amplify neutrophil mediated injury and is involved in both diseases [17]. In addition, MCP-1 is involved in attraction of monocytes and macrophages at the site of inflammation [95] and several studies provided evidence that urinary MCP-1 is elevated in PR3-ANCA and MPO-ANCA vasculitis [75]. It was shown that avacopan, an oral C5a receptor inhibitor, is able to diminish urinary MCP-1 more rapidly than steroids [114]. Thus, urinary MCP-1 might be suitable as biomarker to monitor disease response to treatment. Urinary soluble CD163 (sCD163), shed by monocytes and macrophages, is strongly elevated in cases with active disease and might be a marker of macrophage/monocyte activity, besides its proposed anti-inflammatory properties [96,115]. Chemotaxis is also exerted by soluble Fas [116], which is increased in PR3-ANCA vasculitis. In addition, IL-1ß, IL-6 and the thymus and activation-regulated chemokine (TARC) levels are elevated in both entities [17]. In PR3-ANCA vasculitis, levels of TNF-α, thromboxane A2 (TXA2) and CD14 [117] were increased, while in MPO-ANCA vasculitis the C-C motif chemokine receptor 8 (CCR8) levels was higher and IL-10 expression was lower compared to controls. 

MPO or ROS released by neutrophil degranulation activate complement factors C3 and C5 [118,119]. Neutrophils activated by ANCA also perpetuate activation of complement C3 and its cleavage into C3a and C3b [120], which is observed in both disease entities. Patients with PR3-ANCA vasculitis have an increased expression of the C3 convertase of the alternative complement pathway, namely C3bBbP. On the other hand, since complement receptors also exist in neutrophils, C5a can prime neutrophils and enhance ANCA-induced neutrophil activation [32]. Thus, neutrophils are very closely connected to complement activation. A differential expression pattern among interleukins has been reported, with PR3-ANCA vasculitis associated with IL-10 and IL-32 elevation [82,84]. In contrast, IL-17A and IL-23 are elevated in both, PR3-ANCA and MPO-ANCA vasculitis [93,102]. In PR3-ANCA vasculitis, PR3-ANCA bind strongly to membrane-bound PR3 presented by CD177 [88]. The increased membrane expression of PR3 is dependent upon CD177 expression but not directly linked to circulating PR3 or PR3 gene transcription [87]. Semaphorin 4D (SEMA4D), elevated in both entities, acts as a negative regulator of neutrophil activation and proteolytic cleavage of SEMA4D may amplify neutrophil-mediated inflammatory responses [89]. MIF and matrix metalloproteinase 9 (MMP9), which is known to control the access of monocytes and T cells to the vascular wall [121], are increased in both diseases, while CD14 is involved in the activation of monocytes and neutrophils in PR3-ANCA vasculitis [122]. 

#### 4.1.3. T Cell Activation in PR3-and MPO-ANCA Vasculitis

T cell activation is central in the induction of vasculitis [123]. Th17 effector T cells are shown to be involved in the pathogenesis of ANCA-associated vasculitis [124] and known to affect cytokine levels [125]. In ANCA-associated vasculitis, IL-17A and IL-21 are influenced by Th17 effector T cells and increased in PR3-ANCA vasculitis and elevated IL-17A levels are found in MPO-ANCA vasculitis. Levels of IL-18 and its binding protein (bp) IL-18bp are usually balanced, while in several severe diseases an imbalance has been reported [126]. In both entities, IL-18 and IL-18bp levels are increased compared to controls. The IL-6 receptor (IL-6R) is cleaved by ADAM17 and ADAM10, generating a soluble (sIL-6R) form, which is elevated in both diseases and exert an unknown biologic function [127]. Furthermore, concentrations of soluble IL-6 at baseline correlate with PR3-ANCA titers and increasing concentrations during remission are associated with subsequent disease relapse among rituximab-treated patients [128]. Levels of soluble IL-2R and soluble CD30 are elevated in PR3-ANCA vasculitis. In both diseases, IL-23, TARC and osteopontin, which acts as a structural molecule, humoral factor and cytokine [129], are increased. 

Th1 and Th2 effector T cells are also dysregulated. Th1 cells are overexpressed in ANCA-associated vasculitis, and during acute phases of the disease, it was demonstrated that a higher Th1/Th2 ratio corresponded to higher expression of IFN-γ in the kidneys [130]. Th1 polarization is mediated in ANCA-associated vasculitis by a decrease in CD28, a costimulatory signal which promotes Th2 differentiation [131]. Th1 effector cells promote the secretion of IFN-γ and IgG3, the strongest immunoglobulin subclass in inducing neutrophil activation. This effect reverses during remission, with a polarization toward Th2 response. Patients in remission have higher peripheral counts of Th2 cells, as well as decreased IFN-γ in PBMC supernatant [132]. 

#### 4.1.4. B Cell Activation in PR3-and MPO-ANCA Vasculitis 

B cell stimulation by ANCA-activated neutrophils is associated with an increase in the production of ANCA. BAFF, which is also called B-lymphocyte stimulator (BLyS), is relevant for the development and lifetime of B cells and increases the number of antibody-producing cells. Elevated serum levels of BAFF associate with PR3-and MPO-ANCA vasculitis. Furthermore, elevated levels of BAFF are presented in many B cell driven autoimmune diseases. BAFF levels were also enhanced in patients receiving rituximab therapy [133], further underlining their importance in B cell recovery and antibody production. B cells can also reduce the anti-inflammatory activity of T_reg_ cells and induce the differentiation of effector T cells by secretion of IL-6 and TNF [134]. CD93, a receptor which is expressed during early B cell development [135], is elevated in both entities. In addition, as already stated, TARC levels are elevated in ANCA-associated vasculitis. Moreover, B cell-attracting chemokine 1 (BCA-1), also known as CXC chemokine CXCL13, is an attractant selective for B-lymphocytes [136], and elevated in both diseases. 

#### 4.1.5. Tissue Damage and Repair in PR3-and MPO-ANCA Vasculitis

Both diseases, but especially PR3-ANCA vasculitis, can affect a broad variety of organ systems. Several markers displaying tissue damage and repair processes are dysregulated. Nerve growth factor-ß (NGF-ß) and kidney injury molecule-1 (KIM-1), both associated with inflammatory diseases, are elevated in both diseases [73]. MMPs and tissue inhibitors of metalloproteinase are key elements involved in the formation, remodeling and degradation of matrix protein [137]. As such, key members such as MMP-3, MMP-9 and tissue inhibitor of metalloproteinase (TIMP)-1 are increased in PR3-ANCA and MPO-ANCA vasculitis [73]. Tenascin C (TNC), an extracellular matrix protein, exerts several processes including cell adhesion, but also implicated in cell signaling and gene expression programs [138], was elevated as well as transketolase (TKT), which is an enzyme with implications in the non-oxidative branch of the pentose phosphate pathway [139]. In contrast, levels of platelet derived growth factor-AB (PDGF-AB), involved in the regulation of cellular migration, proliferation and accumulation of extracellular matrix proteins as well as secretion of inflammatory mediators [140], was diminished in both entities. 

#### 4.1.6. Endothelial Injury and Repair in PR3-and MPO-ANCA Vasculitis

Endothelial injury and repair mechanisms are the consequences of active vasculitis. Several upstream mechanisms are implicated in generating ROS, NET formation and local changes. These changes may also explain the high frequency of venous thromboembolic events in patients with ANCA-associated vasculitis [15,46]. 

PR3-ANCA, and to a much lesser extent MPO-ANCA, induces the release of soluble Fms-like tyrosine kinase-1 (sFlt1) from monocytes, resulting in antiangiogenic conditions that interfere with endothelial repair [110]. Intercellular adhesion molecule 1 (ICAM-1), known to be up-regulated in an inflammatory milieu [141], is one of the key factors involved in these processes. Levels of neutrophil gelatinase-associated lipocalin (NGAL), released by various cell types and regulated in diverse processes such as inflammation, ischemia, or infection [142], are increased in both entities. E-selectin is exclusively expressed on endothelial cells, and pro-inflammatory stimuli lead to newly synthesized E-selectin, which is also increased in PR3-ANCA and MPO-ANCA vasculitis. IL-6, one of the major pro-inflammatory cytokines exerting a variety of biologic actions including apoptosis, survival, proliferation, and angiogenesis, is increased in both entities. Clusterin (apolipoprotein J), a ubiquitously expressed glycoprotein having cytoprotective properties [143], is elevated in both diseases. Neo-angiogenesis is necessary to overcome endothelial lesions exerted by active vasculitis. Leucine-rich alpha-2-glycoprotein (Lrg1) is mitogenic to endothelial cells and promotes angiogenesis [144], and is regulated in ANCA-associated vasculitis. S100A8/A9 protein (calprotectin) is capable of exerting pro-inflammatory responses on endothelial cells and thus is elevated in both diseases [145]. 

Another hallmark of vasculitides is the increased frequency of cardiovascular events in comparison to a matched general population [146]. Premature atherosclerosis is one explanation, which has been reported in autoimmunity. An aberrant regulation of the IL-33/soluble suppression of tumorigenesis 2 (sST2) pathway may be one pathophysiological step leading to atherosclerosis [147]. On the other hand, patients with PR3-ANCA and MPO-ANCA vasculitis have decreased levels of plasminogen activator inhibitor-1 (PAI-1), which protects endothelial cells from apoptosis and degradation [73]. 

#### 4.1.7. Role of Proteinase-3

In GPA, associated with PR3-ANCA, dysregulation and hyperactivity of PR3 is relevant in the disease pathogenesis. PR3 synthesis is dysregulated in neutrophils from patients with GPA [148], and higher proportions of neutrophils with significant concentrations of PR3 in the plasma membrane are associated with adverse outcome. The localization of PR3 on the cell surface is mediated by CD18, CD11b, and CD177, a surface protein of neutrophils that binds with high affinity to PR3 [148,149]. This interaction is facilitated through four hydrophobic residues on PR3 that allow it to stably insert into the plasma membrane. This “hydrophobic patch” allows PR3 to bind phosphatidylserine on apoptotic cells, a process facilitated by phospholipid scramblase 1 (PLSCR1) [148,150]. PR3 overexpression on apoptotic neutrophils interferes with efferocytosis exerted by macrophages [151], and GPA particularly presents with an altered localization of the PR3-binding proteins involved in regulating apoptosis, such as annexin-A1, phospholipid scramblase 1, and calreticulin [152,153]. PR3 binds inflammatory microvesicles with high phosphatidylserine concentrations and augments their inflammatory potency (146). The enzymatic activity of membranous PR3 activates secretion of cytokines that stimulate macrophages and dendritic cells [152]. Phosphatidylserine may also function as a receptor for soluble PR3, which may aggravate the vasculitis process. The increased production of PR3 antibodies also predicts relapse in patients treated with rituximab [154]. The crucial role of PR3 is further underlined by the finding that antibody production may precede the development of vasculitis [155]. 

## 5. Differences in Biomarker Expression in PR3-ANCA and MPO-ANCA Vasculitis

Some molecules are elevated or reduced in either PR3-ANCA vasculitis or MPO-ANCA vasculitis or regulated in the same direction. These changes are linked to the pathogenesis of ANCA-associated vasculitis, which may allow for a diagnosis with a set of biomarkers in the future based on the expression levels of those molecules. Table 2 shows the molecules with shared expression between PR3- and MPO-ANCA-associated vasculitis, broken down both in terms of their role in ANCA-associated (AAV) pathogenesis and in typical physiology, and Table 3 shows the molecules that differ between the two vasculitides, PR3- and MPO-ANCA vasculitis. These expression patterns would also help to further provide a tailored approach to treatment of these complex diseases, since it is well established that especially cases with PR3-ANCA vasculitis have a marked relapse risk. Such alterations between both entities were particularly evident for IL-10, which showed significant differences depending on the ANCA serotype, as it is increased in PR3-ANCA and decreased in MPO-ANCA vasculitis [82]. Further confirmatory studies are necessary as a study by Lúdvíksson et al. found no such increase in IL-10 when focusing on patients with GPA and cytoplasmic ANCA positivity [105]. Overall, research should focus on differences among patients with PR3-ANCA and MPO-ANCA vasculitis. 

## 6. Therapeutic Implications of Biomarker Discoveries

### 6.1. TNF-α Inhibitors

TNF-α inhibitors proved effective in several autoimmune disorders. As TNF-α is involved in several pathogenetic steps known to be involved in ANCA-associated vasculitis, several clinical trials have been conducted almost two decades ago. A complex clinical trial included 16 patients each in two independent studies, with study I investigating adjuvant therapy of infliximab in patients receiving cyclophosphamide and prednisolone as part of their remission induction regimen and study II investigating the role of infliximab in patients with persistent disease active despite previous immunosuppressive treatment. In both study arms, 14 patients achieved remission, while three were considered as treatment failures and one patient died. In both groups an expected and significant reduction of the steroid dose could be achieved [156]. Severe infections were observed in 21% of patients, and the rate is comparable to the frequency of severe infectious complications found in rituximab-treated patients [157]. Another single-center study investigated the addition of adalimumab for 3 months in combination with intravenous cyclophosphamide and a reducing course of prednisolone. Out of 14 patients, eleven (78.5%) achieved remission and it was considered that the addition of TNF-α blockade has no significant impact on remission rates while it appeared to be safe [158]. Both trials indicated that the effect of TNF-α inhibition on remission induction is insignificant in the induction of remission. The Wegener’s Granulomatosis Etanercept Trial (WGET) randomized 180 patients with GPA in remission to either etanercept or placebo (both combined with standard therapy). During follow-up, sixty-two of 89 patients in the etanercept group compared with sixty-four of 85 patients in the control group had a sustained remission (69.7% vs. 75.3%). The rate of severe and limited flares was comparable between both treatment arms, so it might be concluded that TNF-α inhibition has a limited effect on the maintenance of remission and thus plays no role in the management of ANCA-associated vasculitis [159].

### 6.2. Interleukin-6 and ANCA-Associated Vasculitis

A paucity of clinical data exists that support the routine use of tocilizumab in the management of ANCA-associated vasculitis. Tocilizumab, a monoclonal antibody targeting the IL-6 receptor, is approved in the management of giant cell arteritis. A recent review of the literature identified 17 cases who received tocilizumab in the management of ANCA-associated vasculitis. A majority (88.2%) achieved remission following IL-6R blockade with tocilizumab [160]. It is likely that more evidence will derive from the current Coronavirus Disease 2019 (COVID-19) pandemic, as tocilizumab is considered as a rather safe alternative to other immunosuppressive measures [161]. 

### 6.3. B-lymphocyte Stimulator (Blys)/B Cell Activating Factor (BAFF) Inhibition

BLyS/BAFF levels are increased in ANCA-associated vasculitis and levels are further increasing one to three months after B cell depleting therapy with rituximab [133]. A phase II trial which randomizes patients to rituximab (2 × 1 g as induction) and belimumab (every week up through week 51) or rituximab (same dosage) and placebo (COMBIVAS) is currently ongoing in patients with PR3-ANCA vasculitis and will look into clinical efficacy but also mechanistic pathways of such a combined approach. A trial of belimumab as adjunctive therapy in the maintenance of remission (BREVAS) terminated recruitment early due to a change in clinical practice and did not provide evidence that patients receiving belimumab would have a reduced relapse risk [162]. Based on these disappointing results, belimumab does not play a role in the maintenance of remission in ANCA-associated vasculitis, but a sequential use of rituximab and belimumab may be used in patients with frequently relapsing disease to block a subsequent increase in BLyS/BAFF levels following B cell depletion with rituximab. Such an approach has provided opposing results in patients with systemic lupus erythematosus with some positive preliminary reports [163], but failed to meet its efficacy endpoint in the recently reported Rituximab and Belimumab for Lupus Nephritis (CALIBRATE) trial [164].

### 6.4. Complement C5a/C5ar Inhibition

The alternative complement pathway is crucially involved in the pathogenesis of ANCA-associated vasculitis [165]. CCX168 (avacopan) treatment in a mouse model dose-dependently reduced the formation of crescents, the key hallmark of renal involvement in patients with ANCA-associated vasculitis. Moreover, hematuria, proteinuria and leukocyturia were significantly reduced in mice receiving CCX168 [166]. Two phase II trials were initiated, CLEAR and CLASSIC. The CLEAR study recruited 67 patients and was performed in three steps. Overall, 23 were assigned to receive standard of care, 22 received avacopan, 20 mg prednisone and standard induction therapy and another 22 received a steroid-free induction therapy. A clinical response was observed numerically more often in the avacopan-treated patients, and in addition a reduction in the urinary albumin/creatinine and MCP-1/creatinine ratios [114]. CLASSIC was considered as a trial to investigate the safety of CCX168 and recruited 42 patients in three arms, with the use of a standard of care (SOC) group as comparator (13 patients), while 13 participants received 10 mg CCX168 twice daily and 16 received 30 mg twice daily on top of SOC. In this trial, CCX168 was found to be safe in both dosing regimens when added to SOC [167]. The phase 3 trial recruited 331 patients to either a steroid-free avacopan group or a standard-of-care group. The trial results were presented recently and the trial met its primary and secondary endpoints [168]. Avacopan will likely be approved as steroid-sparing in the induction of remission of ANCA-associated vasculitis.

### 6.5. Rituximab, the “New Normal”

The success story of rituximab in the management of ANCA-associated vasculitis has been reviewed in detail elsewhere [169]. Both seminal studies, RAVE and RITUXVAS [64,65], showed a non-inferiority of rituximab in the induction of remission compared to a cyclophosphamide-based induction therapy. Notably, in RITUXVAS rituximab was combined with cyclophosphamide. A non-significant increase in malignancies has been reported in RAVE in the rituximab arm. Real-life data from a single-expert center found that malignancy risk following rituximab is not increased during the follow-up period [169,170]. Following a disease relapse, rituximab showed superiority compared to a cyclophosphamide-based induction therapy in achievement of remission [64]. More recently, the induction phase of the Rituximab Vasculitis Maintenance Study (RITAZAREM) trial was reported, and 90% of trial participants achieved remission by 4 months, while only six patients (3.2%) did not achieve disease control [171].

Rituximab not only became the new standard in the induction of treatment, but showed superiority to other agents in the maintenance phase. The Maintenance of Remission using Rituximab in Systemic ANCA-associated Vasculitis (MAINRITSAN) trial found a significant reduction of major relapses during the maintenance period compared to azathioprine without an increase in serious adverse events [172]. Similar results have been reported from the RITAZAREM trial with a superiority of rituximab over azathioprine [171]. There is ongoing concerning debate how long patients should receive maintenance therapy. The recently published MAINRITSAN3 trial recruited patients after completion of an 18-month maintenance regimen and found that prolonged therapy with rituximab significantly reduced the relapse rate, while no increase in rate of serious adverse events was observed [173]. While the efficacy data are reassuring, the side effects of rituximab, including serious infectious complications (progressive multifocal leukencephalopathy, *Pneumocystis jirovecii*, hepatitis B reactivation), long-lasting hypogammaglobulinemia, and late-onset neutropenia need to be taken into account [174]. A figure displaying the mode of action of rituximab and above-mentioned therapeutic options is visualized in Figure 3. 

### 6.6. Developing Preclinical Targets

MPO contributes to oxidative damage involved in the pathogenesis of ANCA-associated vasculitis, suggesting therapeutic utility in MPO inhibition. Antonelou et al. showed that treatment with an MPO inhibitor reduced the production of NETs, ROS, and endothelial cell damage in mice and renal biopsies [175]. Intravenous immunogloblins, previously used to treat other autoimmune vasculitides, reduced the rate of pulmonary hemorrhage and peritoneal NETs in rat models of NETosis [176]. 

## 7. Conclusions

ANCA-associated vasculitis is a systemic autoimmune disease that primarily affects small vessels but may also affect medium-sized vessels. It is currently classified according to the clinical phenotype into three major clinical subtypes: MPA, GPA, and EGPA [1]. Several lines of evidence coming from the genetic background of these diseases, the clinical disease course (association with relapses), disease manifestations (i.e., lung alterations), prognosis (i.e., cardiovascular death) and biomarker expression levels suggest that diagnosis should be made by the respective ANCA serotype, either PR3-ANCA vasculitis or MPO-ANCA vasculitis. 

ANCA-associated vasculitides are complex diseases and research into vasculitides steadily increased during the past decades. Many researchers focused on the immunological pathogenesis and related cytokines in PR3- and MPO-ANCA vasculitis. However, a review focusing on recent findings with involved pathways in etiopathogenesis and associated molecules has not been reported yet. 

The information provided herein might be used clinically to diagnose ANCA-associated vasculitis in unclear cases (especially with non-specific presentation forms). More research is needed to highlight whether these markers are also relevant in the prediction of relapse, which has not been tested in most biomarker studies so far. This review has excluded cases with ANCA-negative vasculitis, a subgroup of cases which is underrepresented in current research, and EGPA, which has a completely different etiopathogenesis when compared to cases with GPA (classically PR3-ANCA positive) or MPA (MPO-ANCA positive). There are some other molecules like IL-10, which have a distinct expression level between PR3-ANCA and MPO-ANCA vasculitis. More research focusing on the ANCA serotype is necessary to provide a panel of potential marker molecules, which are capable of reliably highlighting differences in both entities. In line with our general progress in the field of ANCA-associated vasculitis, the understanding of immunopathogenesis increased during the past decades. Specific biomarkers distinguishing cases with ANCA-associated vasculitis from control cases have been identified and further research in the field is on the way. We are heading towards “precision medicine” in the field of ANCA-associated vasculitis and biomarkers predicting severity of the disease, disease response towards immunosuppression, and relapse risk need to be discovered to allow a tailored treatment. Such examples are the success story of rituximab in the management of ANCA-associated vasculitis and the positive ADVOCATE trial, which will lead to approval of avacopan in the management of these potentially devastating diseases. 

## Figures and Tables

**Figure 1 ijms-21-07319-f001:**
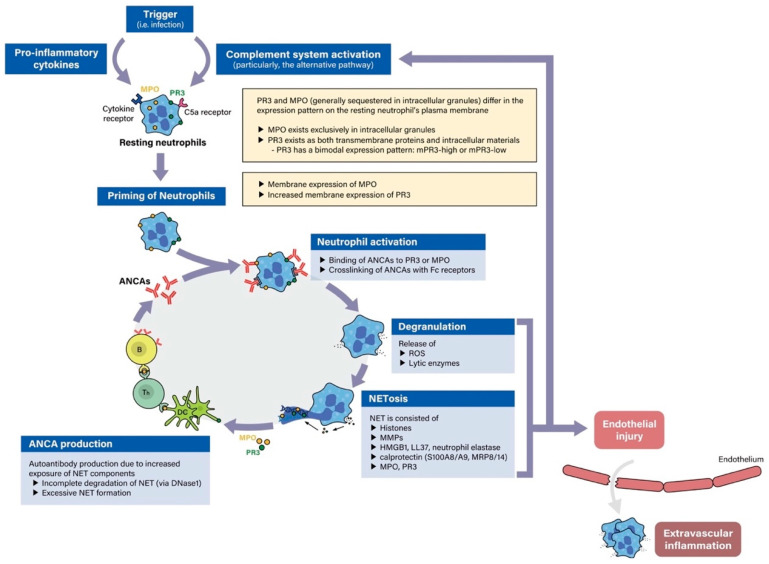
Pathogenesis of anti-neutrophil cytoplasmic antibody (ANCA)-associated vasculitis. An inflammatory trigger leads to increased membranous expression of myeloperoxidase (MPO) and proteinase 3 (PR3) on neutrophils. Binding of ANCAs to PR3 and MPO triggers neutrophil activation, degranulation, neutrophil extracellular trap (NET)osis, which further releases MPO and PR3 to prime ANCAs. Degranulation and NETosis contribute to endothelial injury and complement activation.

**Figure 2 ijms-21-07319-f002:**
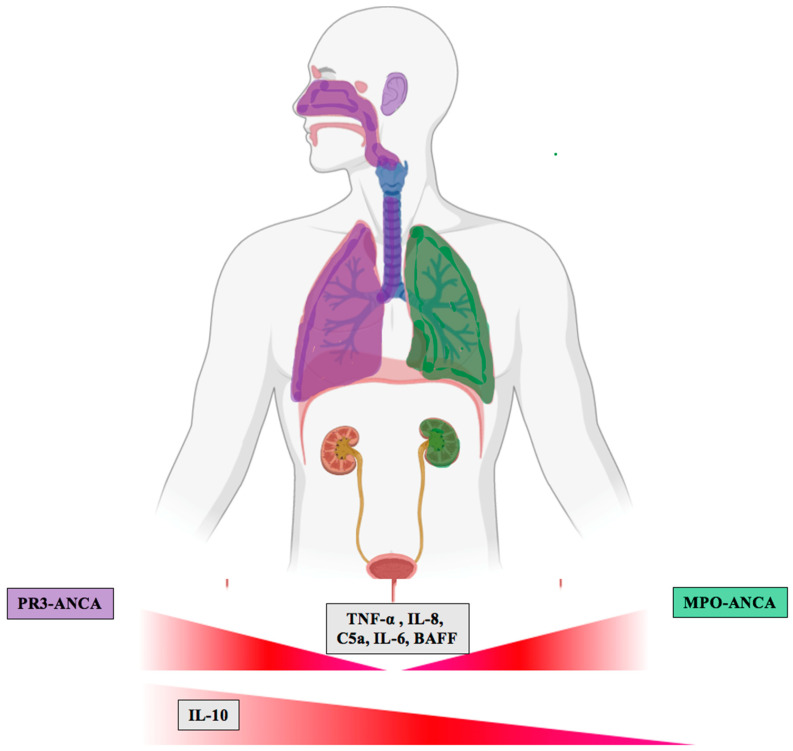
Phenotypes of ANCA serotypes. PR3-ANCA (purple) affects the ear nose and throat (ENT), the upper and lower respiratory tract and the kidneys in around 50–60% of cases, while MPO-ANCA (green) primarily affects the lungs and kidneys. Tumor necrosis factor (TNF)-α, C5a, interleukin (IL)-6, IL-8, and B cell activating factor (BAFF) are elevated in both PR3- and MPO-ANCA. IL-10 is elevated only in PR3-ANCA and is decreased in MPO-ANCA.

**Figure 3 ijms-21-07319-f003:**
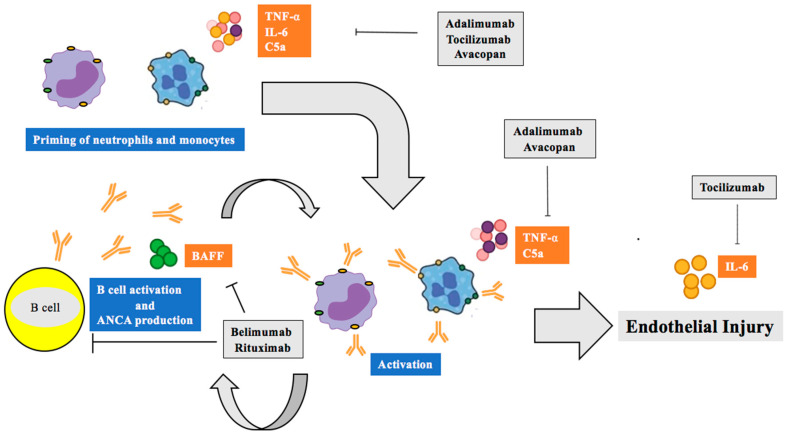
Therapeutic armamentarium of ANCA-associated vasculitis based on disease pathogenesis. TNF-α, IL-6, and alternative complement activation contribute to neutrophil priming, activation, and endothelial injury and are targeted by adalimumab, tocilizumab, and avacopan, respectively. BAFF stimulates B cell activation and is targeted by belimumab. B cells are targeted by rituximab.

**Table 1 ijms-21-07319-t001:** Summary of inflammatory molecules and their expression status in ANCA-associated vasculitis (AAV) classified by ANCA specificity (PR3- or MPO- AAV) and pathogenesis.

		PR3-AAV (vs. Healthy Control)		MPO-AAV (vs. Healthy Control)	
		↑	↓	↑	↓
**Priming**	Neutrophil	TNFα [66,67,68,69,70] IL-6 [71] IL-18 [72,73] IL-2Rα (CD25) [74] C5a [32,75] G-CSF [73] GM-CSF [73,76] HMGB1 [77] MIF [78] ADAM17 [79] α1AT polymers [80]	IL-2Rβ (CD122) [74]	TNFα [66,67,68] IL-6 [71] IL-18 [72,73] IL-2Rα (CD25) [74] C5a [32,75] G-CSF [73,81] GM-CSF [73] HMGB1 [77] MIF [78]	IL-2Rβ (CD122) [74]
	Monocyte	TNFα [66,67,68,69,70] ADAM17 [79]			
**Neutrophil/** **Monocyte** **activation**	Neutrophil	C3a [75] IL-10 [82] IL-17A, IL23 [82,83] IL-32 [84] C3bBbP [85] CD177 [86,87,88] CD14 [70] Semaphorin 4D [89,90] MIF [78] MMP9 [91] sFAS [92]		C3a [75] IL-17A, IL-23 [83,93,94] Semaphorin 4D [89,90] MIF [78] MMP9 [91]	
	Monocyte	MCP-1 [95] Urinary sCD163 [96] TNF-a, IL-1β, IL-6, IL-8, TXA2 [97] CD14 [70] TARC [73]		MCP-1 [95] Urinary sCD163 IL-1β, IL-6, IL-8 [98] TARC [73] CCL18, CCR8 [99]	IL-10 [100]
T cell activation		IL-21 [101] IL-17A, IL-23 [93,102] IL-18BP [73,75] IL-18, sIL-6R, TARC [73] Osteopontin [73,103] sIL2R, sCD30 [104] IFN-γ [105,106]		IL-17A [93,94,102] IL-23 [93,102] IL-18BP [73,75] IL-18, sIL-6R, TARC [73] Osteopontin [73,103]	
B cell activation		BAFF [104] TARC [73] CD93 [91] BCA-1 [41]		BAFF [107,108] TARC [73] CD93 [91] BCA-1 [41]	
Tissue damage and repair		NGFβ [17,73], KIM-1, NGAL, MMP-3, MMP-9, TIMP-1 [73] TNC, TKT [91]	PDGF-AB [73]	NGFβ [17,73], KIM-1, NGAL, MMP-3, MMP-9, TIMP-1 [73] TNC, TKT [91]	PDGF-AB [73]
Endothelial injury and repair		E-selectin, IL-6 [109] NGAL, ICAM-1 [17,73] Clusterin [73], sFlt1 [110] LRG1, S100A8/A9 [91] sST2, IL-33 [111]	PAI-1 [73]	E-selectin, IL-6 [109] NGAL, ICAM-1 [17,73] Clusterin [73], sFlt1 [110] LRG1, S100A8/A9 [91] sST2, IL-33 [111]	PAI-1 [73]

Abbreviations: α1AT (alpha1-antitrypsin), AAV (ANCA-associated vasculitis), ADAM17 (ADAM metallopeptidase domain 17), BAFF (B cell activating factor), BCA-1 (B cell attracting chemokine-1), C3a (complement 3a), C5a (complement 5a), CCL18 (CC-chemokine ligand 18), CCR8 (chemokine C-C motif receptor 8), CD (cluster of differentiation), G-CSF (granulocyte-colony-stimulating factor), GM-CSF (granulocyte macrophage-colony-stimulating factor), HMGB1 (high mobility group box 1), ICAM-1 (intercellular adhesion molecule-1), IFN (interferon), IL (interleukin), KIM-1 (kidney injury molecule-1), LRG1 (leucine rich alpha-2-glycoprotein 1), MCP-1 (monocyte chemotactic protein-1), MIF (macrophage migration inhibitory factor), MMP (matrix metalloproteinase), MPO (myeloperoxidase), NGAL (neutrophil gelatinase-associated lipocalin), NGFß (nerve growth factor-ß), PAI-1 (plasminogen activator inhibitor-1), PDGF-AB (platelet derived growth factor-AB), PR3 (proteinase 3), S100A8/A9 (S100 calcium-binding protein A8/A9), sFlt-1 (soluble Fms-like thyrosinkinase-1), sST2 (soluble growth stimulation expressed gene 2), TARC (thymus and activation regulated chemokine), TIMP-1 (tissue inhibitor of metalloproteinase-1), TNC (tenascin C), TKT (transketolase), TNF (tumor necrosis factor), TXA2 (thromboxane A2).

**Table 2 ijms-21-07319-t002:** Summary of inflammatory molecules with common expression profiles in PR3- and MPO-ANCA vasculitis according to the pathogenesis of AAV.

Biomarker in AAV Pathogenesis	Pathogenesis of AAV (vs. Healthy Controls)	
	↑	↓
**Priming of neutrophils** **(results in ANCA antigen expression on neutrophils’ cell membranes)**	TNF-α [66,67,68] IL-6 [71,73] IL-18 [72,73] IL-2Rα (CD25) [74] C5a [32,75] G-CSF, GM-CSF [73] HMGB1 [77]	IL-2Rβ (CD122) [74]
**Activation of neutrophils**	IL-1β [97,98] C3a [75] Semaphrorin 4D [89,90] MIF [78]	
**Endothelial injury**	E-selectin, IL-6 [109] NGFβ, NGAL, ICAM-1 [17,73] Clusterin [73] sFlt1 [110] sST2, IL-33 [111]	PAI-1 [73]
**Others**	IL-8 [97,98] IL-17, IL-23 [93,94,102] MCP-1 [95] BAFF [101,107] C/EBP-α, C/EBP-β, sFAS [92]	
**Biomarker physiological function**		
**Cytokine**	G-CSF, GM-CSF, IL-6, IL-15, IL-18 [73] Osteopontin [73,103]	
**Chemokine**	BCA-1, IL-8, IP-10, TARC [73]	
**Soluble receptor**	IL-18BP [73,75] sIL-6R, sTNF- RII [73]	
**Tissue damage and repair**	KIM-1, MMP-3, NGFβ, TIMP-1 [73] TNC, CD93, TKT [91] Urinary MCP-1 [75]	PDGF-AB [73]
**Inflammation and vascular injury**	Clusterin, CRP, ESR, ICAM-1, NGAL [73] LRG1, MMP9, S100A8/A9 [91]	PAI-1 [73]
**Others**	Semaphrorin 4D [89,90]	

Abbreviations: AAV (ANCA-associated vasculitis), BAFF (B cell activating factor), BCA-1 (B cell attracting chemokine-1), C/EBP (CCAAT/Enhance-binding protein), C3a (complement 3a), C5a (complement 5a), CD (cluster of differentiation), CRP (C-reactive protein), ESR (erythrocyte sedimentation rate), G-CSF (granulocyte-colony-stimulating factor), GM-CSF (granulocyte macrophage-colony-stimulating factor), HMGB1 (high mobility group box 1), ICAM-1 (intercellular adhesion molecule-1), IL (interleukin), IP-10 (interferon-gamma induced protein 10), KIM-1 (kidney injury molecule-1), LRG1 (leucine rich alpha-2-glycoprotein 1), MCP-1 (monocyte chemotactic protein-1), MIF (macrophage migration inhibitory factor), MMP (matrix metalloproteinase), MPO (myeloperoxidase), NGAL (neutrophil gelatinase-associated lipocalin), NGFß (nerve growth factor-ß), PAI-1 (plasminogen activator inhibitor-1), PDGF-AB (platelet derived growth factor-AB), PR3 (proteinase 3), S100A8/A9 (S100 calcium-binding protein A8/A9), sFlt-1 (soluble Fms-like thyrosinkinase-1), TARC (thymus and activation regulated chemokine), TIMP-1 (tissue inhibitor of metalloproteinase-1), TNC (tenascin C), TKT (transketolase), TNF (tumor necrosis factor).

**Table 3 ijms-21-07319-t003:** Summary of inflammatory molecules with different expression profiles in PR3- and MPO-ANCA vasculitis.

	PR3-ANCA (vs. Healthy Control)		MPO-ANCA (vs. Healthy Control)	
	↑	↓	↑	↓
**Cytokine/cytokine receptors**	IL-10 [82] IL-21 [101] IL-32 [84] sIL-2R, sCD30 [17]			IL-10 [100]
**Chemokine/chemokine receptors**	CD177 [86,87,88] CD14 [70]		CCR8 [99]	
**Complement system**	C3bBbP [85]			
**Others**	ADAM17 [79] TXA2 [97] α1AT polymers [80]			

Abbreviations: α1AT (alpha1-antitrypsin), AAV (ANCA-associated vasculitis), ADAM17 (ADAM metallopeptidase domain 17), CCR8 (Chemokine C-C motif receptor 8), CD (cluster of differentiation), IL (interleukin), MPO (myeloperoxidase), PR3 (proteinase 3), TXA2 (thromboxane A2).

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
