# Peer review of "Immunopathogenesis of ANCA-Associated Vasculitis"

_ijms, 2020, doi:10.3390/ijms21197319_

Round 1

Reviewer 1 Report

This is an up-to-date concise review on pathogenesis of ANCA vasculitis, a complicated issue which is presentes with clarity based on current clinical and investigational data. Certainly the review could benefit from more illustrated figures.

Author Response

Thank you for bringing up this important way to improve our review. We agree and have added two more figures (Figure 1 and Figure 3), which illustrate the relevance of biomarkers to therapy and association between serotype and biomarkers.

Reviewer 2 Report

Today, there are a lot of data available on the pathogenesis and treatments of ANCA-associated vasculitis (AAV) and the authors have revised and discussed some recent works in this field.

I have several issues that need to be addressed before the manuscript should be considered for publication:

  • In my opinion, the proposed model for pathogenesis of AAV is only partial consistent with in vitro/ experimental results and clinical observations. ANCA-associated vasculitis is a multisystem disease characterized by necrotizing vasculitis, glomerulonephritis (GN) and granulomata of the respiratory tract. At sites of vessel wall disruption the acute inflammation and necrosis are involved in induction and promotion of vasculitis, in which inflammatory activity (neutrophils, proteinase 3, NETS, endothelial cells, inflammatory mediators, etc.,), PR3-ANCA and cellular immunity together play a major role.
  • Another area of substantial research in this field pertains to the role of proteinase 3 in the pathogenesis of PR3-ANCA AV. It may be useful to discuss recent studies concerning the role of this molecule.   
  • The presented therapy for PR3- and MPO-ANCA-AV gives a good overview on current treatment strategies, but it highlights new developments of the past year only very briefly.
  • A brief comment on the immunologic differences underlying longitudinal antibody patterns and the associated clinical expression of vasculitis could add to the Discussion.
  • The role of T-cell phenotype (Th1/Th2/Th17) should be discussed
  • Table 2 need to be revised and better explained
  • The authors should use the recommended nomenclature for classification of ANCA-associated vasculitis on the basis of ANCA specificity (e.g., C-, P-, A-ANCA, PR3-ANCA, MPO-ANCA, ANCA negative vasculitis patients).

I suggest to added and discuss other important aspects (i.e., generation of ANCA) and recent publications in this area (vasculitis biomarkers/personalized medicine).

 Other comments:

 Page 3: paragraph 2.1.

“PR3 and MPO are detected in circulation” the authors mean “PR3- and MPO-ANCA”

  • Page 3, paragraph 2.2.

Kessenbrock et al., results should be incorporate in the “neutrophils” paragraph.

Author Response

Reviewer 2

  • In my opinion, the proposed model for pathogenesis of AAV is only partial consistent with in vitro/ experimental results and clinical observations. ANCA-associated vasculitis is a multisystem disease characterized by necrotizing vasculitis, glomerulonephritis (GN) and granulomata of the respiratory tract. At sites of vessel wall disruption the acute inflammation and necrosis are involved in induction and promotion of vasculitis, in which inflammatory activity (neutrophils, proteinase 3, NETS, endothelial cells, inflammatory mediators, etc.,), PR3-ANCA and cellular immunity together play a major role.

Thank you for this comment. We adapted the pathogenesis section accordingly:

# 2.2. Pathogenesis of ANCA-Associated Vasculitis

PR3 has also been shown to reinforce vascular damage in vitro [9]. NET components themselves include PR3 and MPO, and chronic elevation of these enzymes in circulation leads to their recognition by dendritic cells and subsequently T cells and plasma cells as neoantigens [23-25]. The continuous production of PR3-ANCA and MPO-ANCA from lymphocytes results in a vicious cycle of neutrophil hyperactivation, inflammatory activity, and vasculitis. Thus, neutrophils, ANCA elevation, disruption of plasma and T cell tolerance, and overproduction and persistence of NETs together contribute to the pathogenesis of PR3-and MPO-ANCA vasculitis.

  • Another area of substantial research in this field pertains to the role of proteinase 3 in the pathogenesis of PR3-ANCA AV. It may be useful to discuss recent studies concerning the role of this molecule.  

We tried to include the most recent studies investigating the role of PR3 in the pathogenesis of PR3-ANCA vasculitis and briefly discuss them in the context of older studies. We have added the following:

# 2.2. Pathogenesis of ANCA-Associated Vasculitis

RIPK1/3/MLKL-dependent necroptosis induces the release of NETs, which scaffold the alternative complement pathway activation [35].

4.1.7. Role of Proteinase-3

In GPA, associated with PR3-ANCA, dysregulation and hyperactivity of PR3 is relevant in the disease pathogenesis. PR3 synthesis is dysregulated in neutrophils from patients with GPA [149], and higher proportions of neutrophils with significant concentrations of PR3 in the plasma membrane are associated with adverse outcome. The localization of PR3 on the cell surface is mediated by CD18, CD11b, and CD177, a surface protein of neutrophils that binds with high affinity to PR3 [149,150]. This interaction is facilitated through four hydrophobic residues on PR3 that allow it to stably insert into the plasma membrane. This "hydrophobic patch" allows PR3 to bind phosphatidylserine on apoptotic cells, a process facilitated by PLSCR1 [149,151]. PR3 overexpression on apoptotic neutrophils interferes with efferocytosis exerted by macrophages [152], and GPA particularly presents with an altered localization of the PR3-binding proteins involved in regulating apoptosis, such as annexin-A1, phospholipid scramblase 1, and calreticulin [153,154]. PR3 binds inflammatory microvesicles with high phosphatidylserine concentrations and augments their inflammatory potency (146). The enzymatic activity of membranous PR3 activates secretion of cytokines that stimulate macrophages and dendritic cells [153]. Phosphatidylserine may also function as a receptor for soluble PR3, which may aggravate the vasculitis process. The increased production of PR3 antibodies also predicts relapse in patients treated with rituximab [155]. The crucial role of PR3 is further underlined by the finding that antibody production may precede the development of vasculitis [156].

  • The presented therapy for PR3- and MPO-ANCA-AV gives a good overview on current treatment strategies, but it highlights new developments of the past year only very briefly.

We agree and thank you for your comment. We have added the following to expand upon new developments this year, even this is in the early stages (see below). The aim was not to add too much discussions about therapeutic agents as the role of rituximab has been discussed intensively in the past couple of years, and the main avacopan study has not been published so far (ADVOCATE) and the results are only present in Abstract form.  

6.6. Developing preclinical targets

            MPO contributes to oxidative damage involved in the pathogenesis of ANCA-associated vasculitis, suggesting therapeutic utility in MPO inhibition. Antonelou et al. showed that treatment with a MPO inhibitor reduced the producttion of NETs, ROS, and endothelial cell damage in mice and renal biopsies [170]. Intravenous immunogloblins, previously used to treat other autoimmune vasculitides, reduced the rate of pulmonary hemorrhage and peritoneal NETs in rat models of NETosis [171].

  • A brief comment on the immunologic differences underlying longitudinal antibody patterns and the associated clinical expression of vasculitis could add to the Discussion.

We thank you for this point of clarification. We added the following to clarify the distinction between ANCA longitudinal patterns and clinical characteristics/prognoses, with a focus on ANCA negativity and prognoses.

Within GPA, there are few differences in clinical characteristics between patients positive for PR3- or MPO-ANCA. ANCA-negative GPA patients score lower on the Birmingham Vasculitis Activity Score compared to ANCA-positive GPA with a lower frequency of kidney involvement [6]. The degree of ANCA expression may also predict relapse in patients with kidney involvement, regardless of PR3- or MPO-ANCA serotype [48]. ANCA-negativity also predicts longer time to relapse in patients treated with rituximab, as shown by McClure et al. using a 57-patient sample, of which 37% had kidney involvement [49]. However, the value of ANCA-negativity in predicting relapse remains controversial and requires further study. PR3- and MPO-ANCA, their phenotypes, and their associated biomarkers are shown in Figure 2.

PR3-ANCA vasculitis is characterized by a predominant involvement of the upper respiratory tract, and in comparison to MPO-ANCA vasculitis less frequently affects the lower respiratory tract and the kidneys [47]. Both entities present with a similar kidney histology, while in PR3-ANCA vasculitis the number of normal glomeruli and lower amount of interstitial fibrosis is found, explaining the higher rate of kidney recovery in these patients [50,51]. Furthermore, endothelial PR3 internalization leads to apoptosis, while endothelial MPO internalization stimulates intracellular oxidant production [52,53]. Treatment of both MPO- and PR3-ANCA up to date is similar, but more modern treatment strategies and differences in the pathophysiology between both entities and differences in presentation would suggest a more “tailored approach”, i.e. the tempo of kidney function decline. In MPO-ANCA vasculitis, an association between proteinuria and kidney outcome was proposed and these patients may benefit from RAS inhibitors [52].

  • The role of T-cell phenotype (Th1/Th2/Th17) should be discussed

Thank you very much for bringing up this critical aspect of discussion. We have added the following:

     Th1 and Th2 effector T-cells are also dysregulated. Th1 cells are overexpressed in ANCA-associated vasculitis, and during acute phases of the disease, it was demonstrated that a higher Th1/Th2 ratio corresponded to higher expression of IFN-γ in the kidneys [125]. Th1 polarization is mediated in ANCA-associated vasculitis by a decrease of CD28, a costimulatory signal which promotes Th2 differentiation [126]. Th1 effector cells promote the secretion of IFN-γ and IgG3, the strongest immunoglobulin subclass in inducing neutrophil activation. This effect reverses during remission, with a polarization toward Th2 response. Patients in remission have higher peripheral counts of Th2 cells, as well as decreased IFN-γ in PBMC supernatant [127].

  • Table 2 need to be revised and better explained

Thank you for pointing out this critical lack of clarity. We have reevaluated the table and have made the added the following explanation and redesign.

# 5. Differences in Biomarker Expression in PR3-ANCA and MPO-ANCA Vasculitis

These changes are linked to the pathogenesis of ANCA-associated vasculitis, which may allow for a diagnosis with a set of biomarkers in the future based on the expression levels of those molecules. Table 2 shows the molecules with shared expression between PR3- and MPO-ANCA-associated vasculitis, broken down both in terms of their role in AAV pathogenesis and in typical physiology, and Table 3 shows the molecules that differ between the two vasculitides, PR3- and MPO-ANCA vasculitis.

Table 2. Summary of inflammatory molecules with common expression profiles in PR3- and MPO-ANCA vasculitis according to the pathogenesis of AAV.

Biomarker in AAV pathogenesis

Pathogenesis of AAV (vs. healthy controls)

Priming of neutrophils

(results in ANCA antigen expression on neutrophils’ cell membranes)

TNF-α [60-62]

IL-6 [65,67]

IL-18 [66,67]

IL-2Rα (CD25) [68]

C5a [35,69]

G-CSF, GM-CSF [67]

HMGB1 [71]

IL-2Rβ (CD122) [68]

Activation of neutrophils

IL-1β [91,92]

C3a [69]

Semaphrorin 4D [83,84]

MIF [72]

Endothelial injury

E-selectin, IL-6 [103]

NGFβ, NGAL, ICAM-1 [17,67]

Clusterin [67]

sFlt1 [104]

sST2, IL-33 [105]

PAI-1 [67]

Others

IL-8 [91,92]

IL-17, IL-23 [87,88,96]

MCP-1 [89]

BAFF [95,101]

C/EBP-α, C/EBP-β, sFAS [86]

Biomarker physiological function

Cytokine

G-CSF, GM-CSF, IL-6, IL-15, IL-18 [67]

Osteopontin [67,97]

Chemokine

BCA-1, IL-8, IP-10, TARC [67]

Soluble receptor

IL-18BP [67,69]

sIL-6R, sTNF- RII [67]

Tissue damage & repair

KIM-1, MMP-3, NGFβ, TIMP-1 [67]

TNC, CD93, TKT [85]

Urinary MCP-1 [69]

PDGF-AB [67]

Inflammation & vascular injury

Clusterin, CRP, ESR, ICAM-1, NGAL [67]

LRG1, MMP9, S100A8/A9 [85]

PAI-1 [67]

Others

Semaphrorin 4D [83,84]

Abbreviations:...

  • The authors should use the recommended nomenclature for classification of ANCA-associated vasculitis on the basis of ANCA specificity (e.g., C-, P-, A-ANCA, PR3-ANCA, MPO-ANCA, ANCA negative vasculitis patients).

Thank you for bringing this perspective into our work. Unfortunately, because we assembled our data according to the classifications used by the referenced published literature, we will not be able to revise our data without performing undue interpretations on the data. However, we recognize the importance of this suggestion and have added the following to our limitations:

#3.2. Current Classification Criteria

Although classification criteria are currently based on the CHCC nomenclature and are set to change, we classified ANCA-associated vasculitides based on the serotype (PR3- or MPO-ANCA) and based on clinical presentation (GPA or MPA) to maintain consistency with referenced studies in this review.

I suggest to added and discuss other important aspects (i.e., generation of ANCA) and recent publications in this area (vasculitis biomarkers/personalized medicine).

We appreciate the suggestion to expand our discussion. We have added the following:

2.2. Pathogenesis of ANCA-Associated Vasculitis

The initial mechanisms of ANCA generation are not well understood. Chronic nasal colonization with Staphylococcus aureus is associated with relapse in patients with an established diagnosis of GPA, and treatment with trimethoprim-sulfamethoxazole reduced the risk [32]. Treating peripheral blood mononuclear cells (PBMCs) from GPA patients with B cell activating factor (BAFF) and IL-21 increases ANCA production, which is further exacerbated with oligodeoxynucleotides containing CpG motifs, a pathogen-associated immunostimulant [33]. This suggests that hyperactivation of B cells and T cells is involved in initiation of the ANCA production.

 Other comments:

 Page 3: paragraph 2.1.

“PR3 and MPO are detected in circulation” the authors mean “PR3- and MPO-ANCA”

Page 3, paragraph 2.2.

Kessenbrock et al., results should be incorporate in the “neutrophils” paragraph.

We sincerely thank you for these corrections. We have made the requested changes in these locations.

Round 2

Reviewer 2 Report

I see that the paper has improved and can be published.